# Lymphatic Complications Following Sentinel Node Biopsy or Pelvic Lymphadenectomy for Endometrial Cancer

**DOI:** 10.3390/jcm12134540

**Published:** 2023-07-07

**Authors:** Shinichi Terada, Tomohito Tanaka, Hikaru Murakami, Hiromitsu Tsuchihashi, Akihiko Toji, Atsushi Daimon, Shunsuke Miyamoto, Ruri Nishie, Shoko Ueda, Sousuke Hashida, Natsuko Morita, Hiroshi Maruoka, Hiromi Konishi, Yuhei Kogata, Kohei Taniguchi, Kazumasa Komura, Masahide Ohmichi

**Affiliations:** 1Department of Obstetrics and Gynecology, Educational Foundation of Osaka Medical and Pharmaceutical University, Osaka 569-0801, Japan; 2Translational Research Program, Educational Foundation of Osaka Medical and Pharmaceutical University, Osaka 569-0801, Japan

**Keywords:** endometrial cancer, sentinel lymph node, lower-extremity lymphedema

## Abstract

Sentinel node biopsy (SNB) is performed worldwide in patients with endometrial cancer (EC). The aim of this study was to evaluate and compare the occurrence rate of lymphatic complications between SNB and pelvic lymphadenectomy (LND) for EC. The medical records of women who underwent SNB or pelvic LND for EC between September 2012 and April 2022 were assessed. A total of 388 patients were enrolled in the current study. Among them, 201 patients underwent SNB and 187 patients underwent pelvic LND. The occurrence rates of lower-extremity lymphedema (LEL) and pelvic lymphocele (PL) were compared between the patients who underwent SNB and those who underwent pelvic LND. The SNB group had a significantly lower occurrence rate of lower-extremity LEL than the pelvic LND group (2.0% vs. 21.3%, *p* < 0.01). There were no patients who had PL in the SNB group; however, 4 (2.1%) patients in the pelvic LND group had PL. The occurrence rates of lower-extremity LEL and PL were significantly lower in patients who underwent SNB than those who underwent pelvic LND. SNB for EC has a lower risk of lymphatic complications compared to systemic LND.

## 1. Introduction

Lymph node metastasis is one of the most important prognostic factors for endometrial cancer (EC) [1,2,3]. Although there is currently no prospective randomized control trial and the data come from retrospective studies with possible risk of bias, lymphadenectomy (LND) is usually performed to determine lymph node status or to improve prognosis in patients who develop lymph node metastasis [4,5,6,7,8]. However, the occurrence rate of lymph node metastasis is low in patients with low-risk EC [9]. Furthermore, LND does not improve the prognosis in low-risk EC [10,11]. Thus, LND may not be necessary for the treatment of low-risk EC. Moreover, LND may cause post-operative complications and lower-extremity lymphedema (LEL) [12]. LEL is one of the serious postoperative complications caused by LND and is associated with poor quality of life and psychosocial well-being [13,14,15]. It is reported that the occurrence rate of LEL in EC patients ranges from 0 to 50% [16,17,18,19,20].

The sentinel lymph node (SLN) is the first lymph node to receive lymphatic drainage from a primary tumor. Because SLN mapping identifies the primary lymphatic pathway, this technique can increase the detection of lymph node metastases [21,22,23,24], and SLN mapping is not different from LND in lymph node metastasis and recurrence rate [25,26]. It has been reported that sentinel node biopsy (SNB) only reduces the risk of LEL and surgical and post-operative complications [27,28,29,30,31,32,33]. However, the criteria for LEL diagnosis have not been established, and the occurrence rate of LEL is not constant. The objective of this study was to determine the occurrence rate of LEL after surgery, including hysterectomy, bilateral salpingo-oophorectomy with SNB, and pelvic LND without para-aortic LND for EC.

## 2. Materials and Methods

### 2.1. Participants

Among the patients with clinically stage IA endometrioid endometrial cancer of grade 1 or 2 who underwent minimally invasive surgery at the Osaka Medical and Pharmaceutical University in Japan between September 2012 and April 2022, 407 were enrolled in the study. The present study was approved by the institutional review board and obtained written informed consent from the patients (IRB protocol ID: 2012-1120, 2018-082, 2020-087). Patients who declined to participate in the study or had various complications were excluded; 407 of 420 patients were enrolled. All patients had undergone laparoscopic or robot-assisted hysterectomy, bilateral salpingo-oophorectomy with SNB, or pelvic LND. The present study was composed of two prospective cohort studies. In the first study, the patients received SNB with LND for SLN mapping between September 2012 and July 2018. In the second study, the patients received SNB without LND for sentinel node navigation surgery between August 2018 and April 2022. Most surgeries were performed by only one gynecologist (T.T.). All cases were classified using the International Federation of Gynecology and Obstetrics (FIGO) 2008 classification. Patients with synchronous malignant tumor, those who had died, those with unavailable information, those with lower-extremity edema for obvious other reasons, and those who had undergone failed unilateral sentinel node resection were excluded (Figure 1). Patients had not followed up within 3 months after surgery for various reasons were excluded as inadequate medical information. Three patients were excluded from this study because they had additional pelvic and para-aortic LND due to positive SNB. Most patients who had risk of recurrence, including disease progression at stage IB or later, lymphovascular invasion, or high-grade tumor, received chemotherapy as postoperative adjuvant therapy. Some patients who had complications or contraindication against chemotherapy received radiotherapy. They were then reviewed based on their medical records.

### 2.2. Sentinel Node Biopsy

We previously reported on the SLN mapping procedure [34,35,36]. Briefly, the procedure entailed the following: on the day before the operation, 2 mL of 110 MBq 99m-Technetium (99mTc)-labeled tin colloids was injected into the cervix at the 0, 3, 6, and 9 or the 2, 4, 7, and 10 o’clock regions. Lymphoscintigraphy was performed on the same day to identify the regions of the hot spots. On the day of the operation, indigo carmine (2 mg/mL) and indocyanine green (50 µg/mL) were injected into the cervix and fundus of the uterus. SLN was detected via direct viewing using a gamma probe (Navigator GPS; RMD Instruments Inc., Watertown, MA, USA) and a color fluorescence camera (Camera Control Unit JC300, MC Medical Co., Tokyo, Japan for laparoscopy). Then the SLN was resected. When there was no mapping on the hemi-pelvis, side-specific LND was performed. It has been reported that SLN mapping using a cervical injection with combined 99mTc and blue dye and fundus of the uterus injection of blue dye is feasible and accurate in patients with grade 1 endometrial cancer and could have been a reasonable surgical option for this select group of patients; however, SLN mapping failed in approximately 15% of cases [37].

### 2.3. Diagnosis of the Lower-Extremity Lymphedema and Pelvic Lymphocele

The patients with LEL and pelvic lymphocele (PL) were identified via their medical records. The patients underwent physical examinations and LEL was checked for at a postoperative follow-up visit by two gynecologic oncologists (T.T. and S.T.). Patients were examined for PL via ultrasound examination or computerized tomography. Follow-up LEL evaluation and ultrasound examination for PL evaluation was performed at least every 6 months, and CT was performed at least once a year. The patients with obviously different causes of leg edema, such as heart/renal failure, induction due to medication, infection, hypoalbuminemia, and venous thrombosis, were excluded from the study. LEL was graded according to the guidelines proposed by the International Society of Lymphology [38]. The LEL stages were as follows: grade 1, where there is an early accumulation of fluid relatively high in protein content and which subsides with limb elevation; grade 2, where limb elevation alone rarely reduces tissue swelling and where pitting is manifest; grade 3, which encompasses lymphostatic elephantiasis where pitting can be absent and trophic skin changes, such as acanthosis, further deposition of fat, fibrosis, and warty overgrowths, develop.

### 2.4. Statistical Analysis

The statistical analyses were performed using version 14.2.0. of the JMP Pro software program (SAS Institute Japan, Tokyo, Japan). Continuous variables are expressed as the median (inter quartile range) or mean ± standard deviation. The Mann–Whitney U test was used to compare continuous variables, and Fisher’s exact test was used to compare frequencies (non-continuous variables). The incidence of LEL was evaluated using the Kaplan–Meier method. *p* values under 0.05 were considered statistically significant.

## 3. Results

### 3.1. Characteristics

Table 1 shows the characteristics of the patients with EC who underwent SNB or LND. A total of 407 patients met the study criteria. Among the 407 patients, the correct records were unavailable from the medical records in 4 patients, 1 had a synchronous malignant tumor, 4 died, 5 had lower-extremity edema for obviously different reasons, and unilateral sentinel node mapping failed in 5. Therefore, 388 patients were included in the study. Among them, 201 patients underwent SNB and 187 underwent LND. In only one patient could bilateral SNB not be detected, and this patient subsequently underwent full lymphadenectomy.

The mean age (56.4 ± 10.6 vs. 57.4 ± 11.1 years, *p* = 0.35) and the mean body mass index (24.6 ± 5.4 vs. 23.3 ± 6.8, *p* = 0.03) of the patients were not markedly different between the groups. In the SNB group, 199 (99.0%) patients were at stage I of the disease, and 2 (1.0%) were at stage III of the disease. In the LND group, 175 (93.6%) patients were at stage I, and 12 (6.4%) were at stage III. Histologically, in the SNB group, 196 (97.5%) patients had grade 1 or 2 endometrioid carcinoma, 3 (1.5%) had grade 3 endometrioid carcinoma, 1 (0.5%) had serous carcinoma, and 1 (0.5%) had carcinosarcoma. In the LND group, 175 patients (93.6%) had grade 1 or 2 endometrioid carcinoma, 8 (4.3%) had grade 3 endometrioid carcinoma, 5 (2.7%) had serous carcinoma, 1 (0.5%) had carcinosarcoma, and 3 (1.6%) had another histologic type. In the SNB group, 96 (47.8%) patients underwent laparoscopy and 105 (52.2%) underwent robot-assisted surgery. In the LND group, 177 (94.7%) underwent laparoscopy, and 10 (5.3%) underwent robot-assisted surgery. The number of removed lymph nodes in the SNB group was smaller than that of the removed lymph nodes in the LND group (3.0 ± 1.2 vs. 33.8 ± 13, *p* < 0.01). In the SNB group, 22 (10.9%) patients received chemotherapy, and no patient underwent radiotherapy. In the LND group, 37 (19.8%) patients underwent chemotherapy, and 1 (0.6%) underwent radiotherapy as an adjuvant therapy. The median follow-up period was 28 (16–40) months in the SNB group and 75 (58–89) months in the LND group.

Three (1.5%) patients in the SNB group experienced recurrence, compared with 5 (2.7%) in the LND group (*p* = 0.66). The 2-year progression-free survival (PFS) rate was 98.6% in the SNB group and 98.9% in the LND group (*p* = 0.94).

### 3.2. Occurrence Rates of LEL and PL

Table 2 shows the results of tests for the occurrence rates of LEL and PL. The occurrence rate of LEL in the SNB group was lower than that in the LND group (2.0% vs. 21.3%, *p* < 0.01). The median time for LEL development was 10 (5–20) months in the SNB group and 18 (8–37) months in the LND group (*p* = 0.2). In the SNB group, 4 (2.0%) patients had grade 1 LEL, and no patients had grade 2 LEL. Among them, one was more than 80 years old. The number of removed lymph nodes were two or three; there were no factors that could cause LEL with statistically significance. In the LND group, 32 (17.1%) patients had grade 1 LEL and 8 (4.3%) had grade 2 LEL. Grade 3 LEL did not develop in either group. No patient in the SNB group had PL and 4 (2.1%) in the LND group had PL. Among all LEL patients, LEL developed in 16 (36.4%) patients within 1 year, 34 (77.3%) within 3 years, and 41 (93%) within 5 years. The cumulative incidences of LEL in the SNB group at 1 and 3 years were 1.5% and 2.0%, respectively. In the LND group, the rates of LEL that occurred at 1 and 3 years were 7.0% and 16.0%, respectively (Figure 2).

## 4. Discussion

In the current study, the occurrence rates of LEL and PL were significantly lower in the patients who underwent SNB than in those who underwent pelvic LND. The occurrence rates of LEL and PL were 2.0% and 0%, respectively, in the SNB group and 21.3% and 2.1%, respectively, in the LND group.

Table 3 shows the occurrence rate of LEL after SNB and LND for EC mentioned in the previous literature and those observed in our study. In the present study, the occurrence rate of LEL was compared between SNB and pelvic LND without para-aortic LND, and LEL was evaluated via a different method than in the previous literature. In previous studies, the occurrence rate of LEL in patients with EC who underwent SNB ranged from 0–27%, whereas that in those who underwent pelvic with para-aortic LND ranged from 10–40.9% [27,30,31,39]. Thus, previous studies reported that the patients who underwent SNB had a lower occurrence rate of LEL compared to those who underwent LND. These results were in concordance with those of this study. It has been reported in one study that LEL occurred after a median duration of 9.5 months and was diagnosed in 60.3% of patients within 1 year of operation and in 82.1% within 3 years [40]. In this study, LEL was diagnosed in 36.4% of patients within 1 year after surgery and in 77.3% within 3 years.

In the present study, patients were evaluated for LEL via physical examination, the results were graded according to the guidelines of the International Lymphological Society, and the results were not different from those reported for other evaluation methods. Because of the uncertainty in the methods and criteria for LEL diagnosis, the previous reports do not all have a similar occurrence rate of LEL. Diagnostic procedures for LEL include physical examination, ultrasonography, magnetic resonance imaging, magnetic resonance lymphography, computed tomography lymphography, lymphoscintigraphy, and indocyanine green lymphography [41]. It has been reported that the Gynecologic Cancer Lymphedema Questionnaire is useful as a subjective assessment to evaluate patients for LEL [42]. Carlson et al. reported a multicenter prospective study on LEL occurrence after LND for cervical, endometrial, and vaginal cancer. Trained technicians measured the circumference of the bilateral lower limb at 10 cm intervals from the patient’s heel to the groin and calculated the leg volume. The leg volume was evaluated from preoperatively to 24 months postoperatively, and LEL was diagnosed as >10% increase in limb volume. The LEL occurrence rates for cervical, endometrial, and vaginal cancer were 35%, 34%, and 43%, respectively. The LEL occurrence rate peaked between 4–6 weeks after surgery [17].

It has been reported that the occurrence rate of PL after LND was 4–20% for gynecological cancers [19,42,43,44,45], and the patients who underwent SNB had a lower occurrence rate of PL compared to those who underwent LND [27]. These results were in concordance with those of this study. SNB may be able to reduce the risk of the development of PL. In the meta-analysis, laparoscopic lymphadenectomy had lower rate of lymphocele occurrence than laparotomy did. This outcome also depends on the number of resected lymph nodes [46,47].

Lymphatic complications resulting after LND lead to poor quality of life and psychosocial well-being [13,14,15]. Based on our study and other studies, SNB may be able to reduce the occurrence rate of lymphatic complications and maintain the quality of life. Furthermore, patients who undergo minimally invasive surgery, including SNB, had no different prognosis than those who undergo laparotomy [48]. SNB had a high detection rate and low false negative rate in patients with EC [34]. Thus, SNB for low-risk EC may be able to maintain quality of life without resulting in a poor prognosis and be an alternative procedure to LND for EC.

This study is associated with important limitations. First, the sample size was relatively small. Second, the median follow-up period was relatively short in the SNB group and different from the PLD group. A longer follow-up period is required in the SNB group. Third, the diagnosis of LEL may not be objective. Fourth, the accuracy of the diagnosis and grading of LEL may limited because it is evaluated by different persons. Fifth, the patients in the LND group tended to be in more advanced stages of EC than those in the SNB group. Sixth, the present study was a subsequent and not a concurrent comparison, which may decrease the value of the results. Seventh, different surgical techniques and severity of the disease create a bias. Eighth, the study period was 10 years, and the apparatus would be improved during the period; randomization would be needed to during the same period to confirm the results. Therefore, more prospective studies to evaluate the effectiveness and safety of SNB for EC are needed.

## 5. Conclusions

In conclusion, the occurrence rates of LEL and PL were significantly lower in the SNB group than in the LND group. Therefore, SNB for EC can reduce lymphatic complications and also maintain quality of life as compared to systemic LND.

## Figures and Tables

**Figure 1 jcm-12-04540-f001:**
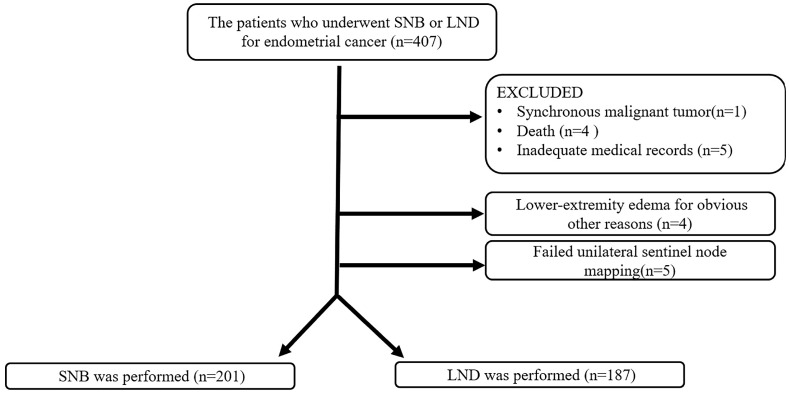
Chart of the study participants who underwent sentinel node biopsy (SNB) or lymphadenectomy (LND). A total of 407 patients met the study criteria. Among the 407 patients, the correct records were unavailable from the medical records in 4 patients, 1 had a synchronous malignant tumor, 4 died, 5 had lower-extremity edema for obviously different reasons, and unilateral sentinel node mapping failed in 5 patients. Therefore, 388 patients were included in the study.

**Figure 2 jcm-12-04540-f002:**
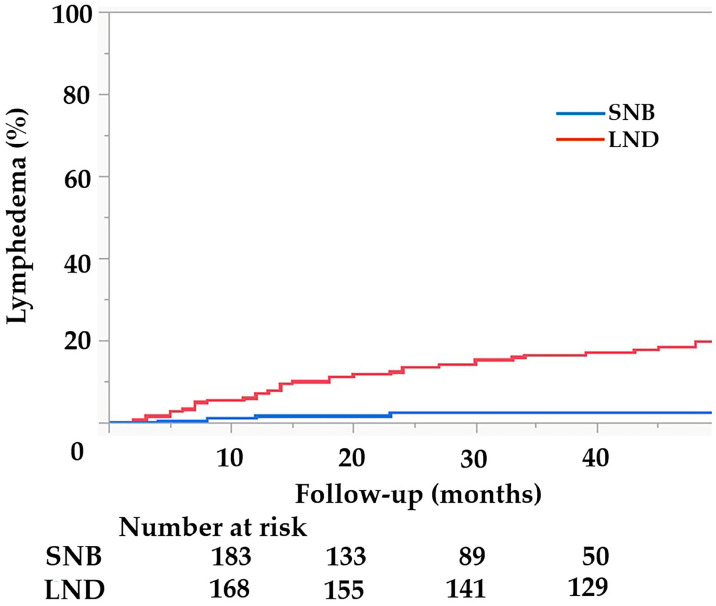
Kaplan–Meier curves of the rate of lymphedema in the sentinel node biopsy (SNB) and lymphadenectomy (LND) groups. The cumulative incidence rates of lower-extremity lymphedema (LEL) in the SNB group at 1 and 3 years were 1.5% and 2.0%, respectively. In the LND group, LEL occurred at 1 and 3 years was 7.0% and 16.0%, respectively.

**Table 1 jcm-12-04540-t001:** Characteristics of the patients who underwent SNB or LND for endometrial cancer.

	SNB	LND	*p* Value
Number of patients	201	187	
Age (years) *	56.4 ± 10.6	57.4 ± 11.1	0.35
BMI *	24.6 ± 5.4	23.3 ± 6.8	0.03
FIGO stage (%)			
I	199 (99.0)	175 (93.6)	<0.01
II	0 (0)	0 (0)	
III	2 (1.0)	12 (6.4)	
Histological type (%)			0.89
Endometrioid grade 1 or 2	196 (97.5)	170 (90.9)	
Endometrioid grade 3	3 (1.5)	8 (4.3)	
Serous carcinoma	1 (0.5)	5 (2.7)	
Carcinosarcoma	1 (0.5)	1 (0.5)	
Other	0	3 (1.6)	
Surgical Approach (%)			<0.01
Laparoscopy	96 (47.8)	177 (94.7)	
Robotic	105 (52.2)	10 (5.3)	
Number of LNs removed *	3.0 ± 1.2	33.8 ± 13	<0.01
Lymph node metastasis (%)	0	3 (1.6)	0.07
Adjuvant therapy (%)			
Chemotherapy	22 (10.9)	37 (19.8)	<0.02
Radiation	0	1 (0.6)	0.5
Follow-up, median months (IQR)	28 (16–40)	73 (58–89)	<0.01

SNB, sentinel node biopsy; LND, lymphadenectomy; BMI, body mass index; FIGO, International. Federation of Gynecology and Obstetrics; LNs, lymph nodes; IQR, interquartile range; * according to an analysis of variance (mean ± standard deviation).

**Table 2 jcm-12-04540-t002:** Occurrence rates of lower-extremity lymphedema and pelvic lymphocele.

	SNB	LND	*p* Value
Number of patients	201	187	
LEL (%)	4 (2.0)	40 (21.3)	<0.01
Grade 1	4 (2.0)	32 (17.1)	<0.01
Grade 2	0	8 (4.3)	<0.01
Grade 3	0	0	
Median months to LEL development (IQR)	10 (5–20)	18 (8–37)	0.2
PL (%)	0	4 (2.1)	

SNB, sentinel node biopsy; LND, lymphadenectomy; LEL, lower-extremity lymphedema; PL, pelvic lymphocele; IQR, interquartile range.

**Table 3 jcm-12-04540-t003:** Literature of LEL after SNB or LND for EC.

Authors	Number of Patients	Follow Up	Method	Incidence of Lower Extremity Lymphedema	Incidence of Pelvic Lymphocele	*p* Value
Geppert et al. (2018) [27]	SNB (*n* = 76) LND* (*n* = 83)	12 months (12–32)	CTC version 3.0	1 (1.3%) 15 (18.1%)	2 (2.6%) 11 (13.3%)	<0.01
Leitao et al. (2019) [30]	SNB (*n* = 180) LND** (*n* = 352)	63 months (44–101) 93 months (44–131)	LELPRO survey	49 (27.2%) 144 (40.9%)	NR	<0.01
Accorsi et al. (2018) [31]	SNB (*n* = 61) LND** (*n* = 89)	90 days	MSKCCSSEGS	0 (0%) 9 (10.1%)	NR	<0.01
Glaser et al. (2020) [39]	SNB (*n* = 127) LND** (*n* = 41)	25 months (21–29) 51 months (32–72)	LEL screening questions	33 (26.0%) 41 (39.0%)	NR	<0.01
Our study	SNB (*n* = 201) LND*** (*n* = 247)	28 months (16–40) 73 months (49–94)	ISL classification	4 (2.0%) 40 (21.3%)	0 (0%) 4 (2.1%)	<0.01

SNB, sentinel node biopsy; LND, lymphadenectomy; LEL, lower-extremity lymphedema; CTC, common toxicity criteria; PRO, patient-reported outcome; MSKCCSSEGS, Memorial Sloan Kettering Cancer Center surgical secondary events grading system; ISL, International Society of Lymphology; LND*, pelvic and para-aortic LND; LND**, pelvic with or without para-aortic LND; LND***, pelvic LND.

## Data Availability

The datasets used and analyzed during the current study are available from the corresponding author on reasonable request.

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
