# Peer review of "Lymphatic Complications Following Sentinel Node Biopsy or Pelvic Lymphadenectomy for Endometrial Cancer"

_jcm, 2023, doi:10.3390/jcm12134540_

Round 1

Reviewer 1 Report

I read with interesting with your work however there are some points that probably should be addressed before publication 

1 There is a total of 11 patients with stage Ia grade 3 and only one of them receiving post operative radiation as adjuvant therapy, please explain the reason ?

2 59 of your patients receiving postoperative chemotherapy ,however there's only 11 in your cohort having histology other than endometroid type would you please address the reason for chemotherapy ?

3 Interesting to see there are 2%of patients developing LEL in the SNB cohort any particular risk factors for these 4 patients who develop LEL after merely SNB?

Author Response

We appreciate the time and effort of the editor and referees in reviewing our manuscript. We have addressed all the issues indicated in the review report and hope that the revised version meets the journal’s requirements for publication.

Response to Comments from Reviewer 2:

Comment 1:

I read with interesting with your work however there are some points that probably should be addressed before publication. There is a total of 11 patients with stage Ia grade 3 and only one of them receiving post operative radiation as adjuvant therapy, please explain the reason?

Response:

Although adjuvant radiotherapy is a standard treatment in many countries, chemotherapy is widely performed as adjuvant therapy for endometrial cancer patients with a risk for recurrence in Japan. One patient who had complications of contraindication against chemotherapy received radiotherapy.

According to your suggestion, we added the sentences as follows.

Most patients who had risk of recurrence, including disease of stage IB or more, lymphovascular invasion, or high-grade tumor, received chemotherapy as postoperative adjuvant therapy. Some patients who had complications of contraindication against chemotherapy received radiotherapy.  (page2, line 76-79)

Comment 2:

59 of your patients receiving postoperative chemotherapy, however there's only 11 in your cohort having histology other than endometroid type would you please address the reason for chemotherapy?

Response:

As you pointed out, 59 patients received adjuvant chemotherapy in this study. In our study, adjuvant therapy was performed to the endometrial cancer patients at risk of recurrence, who had disease of stage IB or more, lymphovascular invasion, or high-grade tumors. Therefore, 59 patients received postoperative adjuvant chemotherapy.

According to your seggestions, we added the sentences as follows.

 Most patients who had risk of recurrence, including disease of stage IB or more, lymphovascular invasion, or high-grade tumor, received chemotherapy as postoperative adjuvant therapy. (page2, line76-78)

Comment 3:

 Interesting to see there are 2%of patients developing LEL in the SNB cohort any particular risk factors for these 4 patients who develop LEL after merely SNB?

Response:

As you pointed out, there were 4 patients developing LEL in the SNB group. one was older than 80 years-old. The number of removed lymph nodes were two or three; there were no factors that could cause LEL with statistically significant.  

According to your suggestions we added the sentences as follows.

Among them, one was older than 80 years-old. The number of removed lymph nodes were two or three; there were no factors that could cause LEL with statistically significant. (page5, line 167-169)

Reviewer 2 Report

This paper dealt with non-randomized, exploratory prospective cohort studies in sequential phase 2 setting concerning to sentinel lymphadenectomy and total pelvic lymphadenectomy in patients with clinical stage IA of endometrial cancer using laparoscopy or robotic surgery.

The present study is not clear to determine whether prospective or retrospective.  Considering the study period was 10 years (2012-2022), the apparatus would be improved during the period. 

Randomization would be needed to during the same period to confirm the results. 

Method

In this item, referring the previous paper by Abu-Rustem, et al., author should mention that SLN mapping using a cervical injection with combined 99mTc and blue dye was feasible and accurate in patients with grade 1 endometrial cancer and may be a reasonable surgical option for this select group of patients, however, approximately 15% of cases with failed SLN mapping. (Abu-Rustem NR, et al.2009, Gynecol oncol. 113.163:doi: 01/j.ygyno.2009.01.003)

Page 2, line 69 to76: These detail explanation and Fig. 1 would be written in the results.

Page 3, Line 84 to 94: Concerning to the injection points, authors should be mentioned that cervical points and fundal points were most appropriate for endometrial cancer as well by Abu-Rsutem’s paper for easy to understand of readers.

Results

Author should mention recurrent rate and prognosis (PFS and/or OS) of minimal surgery in this study.

Discussion

Line 178: In the previous study----  In the previous studies-----

Line 209   It is reported----- It has reported-------

Author Response

We appreciate the time and effort of the editor and referees in reviewing our manuscript. We have addressed all the issues indicated in the review report and hope that the revised version meets the journal’s requirements for publication.

Response to Comments from Reviewer 3:

Comment 1:

This paper dealt with non-randomized, exploratory prospective cohort studies in sequential phase 2 setting concerning to sentinel lymphadenectomy and total pelvic lymphadenectomy in patients with clinical stage IA of endometrial cancer using laparoscopy or robotic surgery. The present study is not clear to determine whether prospective or retrospective.  

Response:

As you pointed out, this study may be not clear whether prospective or retrospective. This study is prospective cohort study, because almost all patients who were eligible for this study were enrolled.

According to your suggestion, we added the sentences as follows.

407 of 420 patients were enrolled. (page2, line62-63)

Comment 2

Considering the study period was 10 years (2012-2022), the apparatus would be improved during the period. Randomization would be needed to during the same period to confirm the results.

Response:

As you mentioned, this study could not be randomized. We added the limitation as follows.

Eighth, the study period was 10 year and the apparatus would be improved during the period; randomization would be needed to during the same period to confirm the results. (page7, line 247-249)

Comment 3:

Method

In this item, referring the previous paper by Abu-Rustem, et al., author should mention that SLN mapping using a cervical injection with combined 99mTc and blue dye was feasible and accurate in patients with grade 1 endometrial cancer and may be a reasonable surgical option for this select group of patients, however, approximately 15% of cases with failed SLN mapping. (Abu-Rustem NR, et al.2009, Gynecol oncol. 113.163:doi: 01/j.ygyno.2009.01.003)

Response:

Based on your suggestion, we have added sentences as follows.

It has reported that SLN mapping using a cervical injection with combined 99mTc and blue dye and fundus of the uterus injection of blue dye was feasible and accurate in patients with grade 1 endometrial cancer and may be a reasonable surgical option for this select group of patients, however, approximately 15% of cases with failed SLN mapping.

(page2, line 91-95)

Comment 4:

Page 2, line 69 to76: These detail explanation and Fig. 1 would be written in the results.

Response:

According to your suggestion, we added the sentences about figure 1 in results section as follows.

Table 1 shows the characteristics of the patients with EC who underwent SNB or LND. A total of 407 patients met the study criteria. Among the 407 patients, the correct records were unavailable from the medical records in 4 patients, and 1 had synchronous malignant tumor, 4 died, 5 had lower-extremity edema for obvious other reasons, and 5 failed unilateral sentinel node mapping. Therefore, 388 patients were included in the study. (page3-4, line 129-133)

Comment 5:

Page 3, Line 84 to 94: Concerning to the injection points, authors should be mentioned that cervical points and fundal points were most appropriate for endometrial cancer as well by Abu-Rsutem’s paper for easy to understand of readers.

Response:

According to your suggestions, we added the sentences as follows.

It has reported that SLN mapping using a cervical injection with combined 99mTc and blue dye and fundus of the uterus injection of blue dye was feasible and accurate in patients with grade 1 endometrial cancer and may be a reasonable surgical option for this select group of patients, however, approximately 15% of cases with failed SLN mapping. (page2, line 91-95)

Comment 6:

Results

Author should mention recurrent rate and prognosis (PFS and/or OS) of minimal surgery in this study.

Response:

According to your suggestions, we added the sentences about prognosis as follows.

Three (1.5%) patients experienced recurrence in the SNB, compared with 5 (2.7%) in the LND group (p = 0.66). The 2-year progression -free survival (PFS) rate was 98.6% in the SNB group and 98.9% in the LND group (p = 0.94). (page5, line 159-161)

Comment 7:

Discussion

Line 178: In the previous study----  In the previous studies-----

Response:

According to your suggestion, we revised the sentence.  (page 6, line 195)

Comment 8:

Line 209   It is reported----- It has reported-------

Response:

According to your suggestion, we revised the sentence. (page 7, line 225)